# The Brain Pre-Metastatic Niche: Biological and Technical Advancements

**DOI:** 10.3390/ijms241210055

**Published:** 2023-06-13

**Authors:** Maximilian Geissler, Weiyi Jia, Emine Nisanur Kiraz, Ida Kulacz, Xiao Liu, Adrian Rombach, Vincent Prinz, Daniel Jussen, Konstantinos D. Kokkaliaris, Hind Medyouf, Lisa Sevenich, Marcus Czabanka, Thomas Broggini

**Affiliations:** 1Department of Neurosurgery, University Hospital, Goethe-University, 60528 Frankfurt am Main, Germany; 2Dr. Senckenberg Institute of Pathology, University Hospital Frankfurt, 60528 Frankfurt am Main, Germany; 3German Cancer Consortium (DKTK), Partner Site Frankfurt/Mainz, 60528 Frankfurt am Main, Germany; 4Frankfurt Cancer Institute (FCI), Goethe University Frankfurt, 60528 Frankfurt am Main, Germany; 5Georg-Speyer-Haus, Institute for Tumor Biology and Experimental Therapy, 60528 Frankfurt am Main, Germany; 6German Cancer Research Center (DKFZ), 69120 Heidelberg, Germany

**Keywords:** brain metastasis, pre-metastatic niche, metastatic niche, brain pre-metastatic niche, vascular niche, single-cell sequencing, transcriptomics, advanced imaging

## Abstract

Metastasis, particularly brain metastasis, continues to puzzle researchers to this day, and exploring its molecular basis promises to break ground in developing new strategies for combatting this deadly cancer. In recent years, the research focus has shifted toward the earliest steps in the formation of metastasis. In this regard, significant progress has been achieved in understanding how the primary tumor affects distant organ sites before the arrival of tumor cells. The term pre-metastatic niche was introduced for this concept and encompasses all influences on sites of future metastases, ranging from immunological modulation and ECM remodeling to the softening of the blood–brain barrier. The mechanisms governing the spread of metastasis to the brain remain elusive. However, we begin to understand these processes by looking at the earliest steps in the formation of metastasis. This review aims to present recent findings on the brain pre-metastatic niche and to discuss existing and emerging methods to further explore the field. We begin by giving an overview of the pre-metastatic and metastatic niches in general before focusing on their manifestations in the brain. To conclude, we reflect on the methods usually employed in this field of research and discuss novel approaches in imaging and sequencing.

## 1. Introduction

Cancer poses the greatest burden of any disease clinically, socially, and economically and is the second cause of death worldwide. Metastasis, the spread of cancer cells from the primary tumor to distant organs, is the leading cause of death in cancer patients [1]. Brain metastasis is a devastating complication of cancer that affects approximately 20% of all cancer patients [2], yet it eludes effective treatment and has been extraordinarily hard to decipher [3]. In recent years, metastasis research in general and brain metastasis research in particular have seen productive surges fueled by advances in both pre-clinical and clinical techniques [4,5,6].

For example, the recognition of the microenvironment as a key factor in the development and progression of cancer metastasis forms the concept of the metastatic niche (MN) [7]. Another key insight was the discovery of the pre-metastatic niche (PMN)—this proposed microenvironment is established by the primary tumor at a distance, ripening the sites of future metastasis before tumor cells reach the final steps of extravasation, migration, and invasion. The PMN is established via the secretion of exosomes, cytokines, growth factors, and other signaling molecules. These factors modify the composition of the extracellular matrix in the target organs via the recruitment of immune and stromal cells. This deliberately generated microenvironment provides a soil in distant organs to accept circulating tumor cells and support the survival, growth, and colonization of cancer cells. The concept has opened up new possibilities for understanding some of the most intriguing questions in cancer biology [8].

Studying the brain PMN has developed into an area of active research with promising discoveries. Our review aims to provide an overview of recent understandings in this area and to discuss possible experimental approaches, leading to more insights in the future.

## 2. General Concepts of the Pre-Metastatic Niche

Past metastasis research focused on the six basic steps of the metastatic cascade: primary tumor growth, tumor intravasation, tumor cell dissemination, tumor cell extravasation, tumor cell adaption/dormancy, and a distant tumor in the metastasis process. In hindsight, this provided relatively simple description of the actual processes [9]. The prevailing model did not account for complex signaling by primary tumors and the interplay between pioneering micrometastases and the extracellular matrix, including the multitude of angiogenetic factors [10].

This dogma was questioned when the initiation of a pre-metastatic niche was observed for the first time. Kaplan et al. showed a mechanism through which bone-marrow-derived stem cells (BMDCs) clustered in future sites of metastasis in the lung and upregulated local fibronectin production to promote the adhesion of circulating tumor cells [11]. Through this mechanism, a fertile soil can be generated by a primary tumor to accept its seeds—this reflects a refinement of the seed and soil concept that was famously proposed by Paget and Fuchs in 1889 [12]. Since then, a number of theoretical PMN mechanisms have been explored for their ability to modify organ sites from a distance (Figure 1). They range from preparing the extracellular matrix by upregulating pro-adhesive factors to the early initiation of angiogenesis [13,14]. Discoveries concerning endothelial cells consistently emerge in the PMN literature [8,15,16,17,18,19]. The endothelial wall is the first barrier to circulating tumor cells blocking intra and extravasation. Hence, an important role of the endothelium in early metastasis is to be expected. The remodeling of endothelial cells and the upregulation of cell adhesion molecules by melanoma and lung carcinoma cells was found to be mediated via hijacking the endothelial signal transducer activator of transcription 3 (STAT3) [20]. Similarly, endothelial adhesion seems to be promoted in estrogen-receptor-negative breast cancer cells via E-selectin [21]. Endothelial cells are also modified through the upregulation of the Notch1 receptor, promoting senescence and inflammation, which are conducive to metastasis [22]. In a spontaneous murine breast cancer model, pre-metastatic lung tissue experienced an influx of immature myeloid cells, which subsequently remodeled the endothelium via matrix metalloproteinase 9 (MMP9) and promoted metastasis—a phenotype that could be reverted via MMP9 deletion [17].

Over the last two decades, the importance of the immune system in cancer was established. This sprouted a fertile research field known as immuno-oncology [9,10,23,24]. For example, the mesenchymal–epithelial–transition is seen as a crucial step in providing a pro-invasive microenvironment and is controlled by a number of immunological factors, including transforming growth factor (TGF)-β [6], which is widely characterized in a pre-metastatic setting [25,26,27]. Immunological modulations, the recruitment of immunosuppressive cells and the ambiguous plasticity of neutrophils have attracted the interests of metastasis researchers in general and the field of PMN scientists in particular [28,29]. In fact, immunological modulation can move in both suppressive and activating directions as cancer makes twofold use of the immune system [30]. Immune escape phenomena are well known in the tumor microenvironment in which regulatory T-cells (T_reg_) and tumor-associated macrophages (TAM) suppress CD4^+^ and CD8^+^ T-cell activation [30,31,32,33]. TAMs create an immuno-suppressive environment via the expression of anti-inflammatory cytokines (TGFb1, IL1 receptor antagonist and IL10) [34], while tumor inflammatory monocytes (TIMs) promote a pro-inflammatory milieu (via IL1β and TNF) [35,36]. Interestingly, profiling TIMs across different metastatic sites and primary tumors revealed significant changes in their pro-inflammatory signatures depending on the origin of the tumor. Functional assays linked the presence of TIMs and TAMs with the exhaustion of cytotoxic and helper T cells in the metastatic site via the CCL20-CCR6 axis. Blocking this axis reverted immunosuppression, reduced T-cell exhaustion and delayed the progression of metastasis in mouse models [35], illustrating the value of modulating the immune microenvironment as an alternative therapeutic strategy to indirectly suppress tumor cell expansion. Single-cell profiling of bone metastasis samples from prostate cancer patients revealed significant changes in the transcriptional signatures of myeloid cells located at the metastatic versus distal vertebral sites [35]. Recently, a pre-metastatic immune modulation of circulating tumor cells was discussed, but the field still lacks conclusive results regarding those effects in the PMN or even the MN [37]. 

The mechanisms through which specific organs are targeted by primary tumors and how they form niches remain under investigation. Some of the most promising candidates are tumor-derived exosomes, which carry surface proteins with organotropic signatures [38,39]. This allows organotropism in a range of different cancer types via variations in integrin expression in endothelial cells [40]. Moreover, exosomes modulate the initial and subsequent tumor microenvironments of the primary and the metastatic tumors, respectively [41]. Exosomes from pancreatic ductal adenocarcinomas (PDACs) were found to regulate Kupffer cells in the liver, increasing pro-invasive fibronectin production [25]. The upregulation of Rab-proteins (notably *RAB1A*, *RAB5B*, *RAB7* and *RAB27A*), which have implicit functions in membrane trafficking and exosome formation, can be found in a range of cancers [42,43,44]. 

Pro-angiogenic modulation in different tumors and organs by exosomal non-coding RNA (ncRNA) is well documented [14,45,46,47]. Recently, the influence of ncRNA in metastasis and its effect on distant organ sites has been introduced. Through the direct genetic manipulation of target cells and easy packaging in exosomes, ncRNA acts as an early effector and modulator of the pre-metastatic niche [18,48,49,50,51]. The organ specificity of tumor-derived exosomal ncRNA promises to improve targeted therapy and will identify novel therapeutic targets [52,53].

## 3. Recent Progress in Understanding the Brain Pre-Metastatic Niche

Empirical evidence for a biological niche that functionally predetermines metastatic seeding in the brain is scarce. One of the most important aspects that sets brain metastasis formation apart from other organ sites is the exclusive entry through the BBB, as there is no lymphatic system capable of introducing metastases in the brain [3,54,55]. Therefore the brain endothelium promises to be the main target for further research into this topic (Figure 2), and its possible role in cerebral metastases has been long appreciated [56].

An early event in brain metastatic colonization represents the activation of platelets through the von Willebrand factor (VWF) produced by platelets and cerebral endothelial cells. VWF fibers promote platelet clustering and support tumor cell arrest in vessels before extravasation. Interestingly, accumulations of VWF fibers were observed at similar extents in vessels in brain metastatic and perimetastatic regions, as well as in cerebral vessels in tumor-bearing mice without established brain metastasis, indicating a role in premetastatic niche formation [57]. Moreover, glial cells, in particular astrocytes and microglia, are active mediators of cerebral endothelial cell remodeling and are recruited to sites of future metastasis, where they undergo phenotypic switching. In melanoma brain metastasis, astrocyte recruitment involves the upregulation of neuroinflammatory and astrogliotic genes (most notably *CCL2*, *CCl17*, *CXCL10*, *IL-23* and *GFAP*), leading to the formation of a glia scar and BBB disruption—a process well documented in other neuroinflammatory diseases [58,59,60,61,62] (Figure 2). Rodrigues et al. demonstrated that brain tropic breast cancer cells (231-BrT) shed exosomes containing high levels of CEMIP (cell migration-inducing and hyaluronan-binding protein), which increased cytokine production in the microglia with the effect of neuroinflammation and increased the leakiness of the neurovasculature. Although an effect on metastasis formation could be seen in vivo after pre-treatment with exosomes, cellular knockdown of CEMIP also abrogated metastasis. Hence, one could argue that the tumor-growth-suppressing functions of CEMIP significantly confounded the findings [63] (Figure 2). In overt breast cancer metastasis, reactive SIP3 (sphingosine-1 phosphate receptor 3) signaling from astrocytes in the neuroinflammatory response has been shown to regulate the vascular–tumor interface in overt breast cancer metastasis, highlighting a sustained importance of neuroinflammation throughout the metastatic process [64]. Neuroinflammation and glial activation were long regarded as host defense responses leading to tumor cell eradication. However, it is increasingly recognized that the interaction between metastatic tumor cells and microglia in the vascular niche leads to microglial reprogramming and functional cooption. This results in tumor-promoting functions across the stages of metastatic colonization, including vascular remodeling, tumor extravasation and outgrowth [65,66,67,68]. Likewise, functional cooption and the blockading of astrocytes’ anti-cancer functions by metastatic tumor cells at the neurovascular interface have been described and associated with increased brain metastatic potential and the instigation of neuroinflammatory responses [69,70,71]. Accordingly, neuroinflammatory responses have been documented in a number of human brain metastasis samples [72,73].

Paracrine secretions have been found to be sources of pre-metastatic signaling. For example, neural progenitor cells (NPC) were transformed into astrocytes in a co-culture with breast cancer cells via paracrine communication through transforming growth factor-β (TGF-β) family factor BMP-2 (bone morphogenic protein) [74] (Figure 2). As in vivo experiments require a strong ethical justification, co-culture data provide a good starting point to gauge individual research objectives and establish more elaborate animal experiments. In another case, the placental growth factor (PLGF) secreted by small-cell lung cancer (SCLC) has been shown to mediate the disassembly of brain endothelial tight-junctions and promote brain metastasis both in vitro and in patient samples, but the study lacks the in vivo data vital to drawing conclusions with respect to the PMN [75] (Figure 2). The role of vascular endothelial growth factor (VEGF) in metastasis is generally well understood, and its disruptive effects on the BBB have been documented both in vivo and in vitro [76,77,78]. VEGF-A-containing exosomes secreted by glioblastoma cells have been shown to open the BBB and promote angiogenesis [79], suggesting a similar role in metastasizing cancer (Figure 2). Vascular leakiness as an influence of estrogenic regulation has also been proposed in breast-cancer-associated metastasis but lacks experimental confirmation [80].

Finally, exosomal ncRNA has been shown to have a direct influence on future sites of metastasis. Its role in brain metastasis is mostly unexplored, but a few promising insights have been gained already. ncRNA was shown to downregulate the expression of tight junction proteins and promote cerebral metastasis in a murine NSCLC (non-small-cell lung cancer) model and was similarly proposed in an in vitro model of human breast cancer cells [81,82]. Fong et al. showed that the miR-122 secreted by breast cancer cells alters glucose metabolism in astrocytes, which favors the metabolism of metastatic cells (Figure 2). They were able to isolate the exosomes via gradient separation from a number of different breast cancer cell lines and conduct mechanistic experiments in vitro. By pre-conditioning mice with exosomes and through the subsequent injection of tumor cells, a strong effect on metastatic growth could be shown in vivo [83]. This study exemplifies the necessity for an observation of the proposed mechanisms isolated from confounders, such as circulating tumor cells, to convincingly show a real pre-metastatic effect. In another investigation on breast cancer exosomes by the same group which used comparable methodology, miR-105 was able to destroy endothelial tight junctions, and its upregulation led to more brain metastasis in a murine model [19] (Figure 2). This multitude of effects implies that more significant discoveries lie ahead for studying exosomal RNA in brain metastasis. Specific RNA-targeted therapeutics have also been discussed in the context of brain metastasis, as they have been for primary tumor RNA therapeutics, but to date, no suitable candidates could be discovered [84,85].

## 4. Differences between the PMN and MN

The pre-metastatic and metastatic niches are parts of the early phases of metastasis formation and although one precedes the other conceptually, they share considerable overlap in both mechanisms and effectors (Table 1) [7,8]. It can be assumed that the underlying molecular processes occur simultaneously and interdependently.

However, the pre-metastatic niche refers to the microenvironment that is created in a distant organ by the primary tumor prior to cancer cell infiltration. It provides a supportive environment by inducing vascular disruption and promoting extravasation, angiogenesis, inflammation and the early remodeling of the extracellular matrix via the recruitment of stromal cells [8].

The transition to the metastatic niche is fluent and begins after circulating cancer cells arrive in this pre-established tumor microenvironment. The establishment of the metastatic niche can be characterized by pro-survival signals of the pioneering micro-metastatic cells and the proliferation and initiation of angiogenesis. The goal of the first micro-metastasis is to foster self-survival and form stable growth and invasion [7].

## 5. Discussion

### 5.1. Experimentally Defining the PMN

The most obvious challenge in defining any characteristics of a hypothetical pre-metastatic niche is to capture the right time in the metastatic cascade in order to identify a causal link between a secreted factor from the primary tumor and the colonization of a specific organ [55]. Different approaches have been used to address this causality. One of the earliest methods described involves flow-cytometric measurements of GFP-tagged BMDCs and DsRed-tagged B16 murine melanoma cells at different time points in the lung lysates of mice previously injected orthotopically with B16 cells. The dynamic changes in cell type in the microenvironment show an influx of BMDCs days before the arrival of tumor cells, suggesting the formation of hospitable pre-metastatic conditions [11]. This method can show the dynamic composition of a proposed niche and clearly delineates its temporal component, yet it requires strong candidate factors previously identified for a successful experiment and does not necessarily prove a causal relationship. Moreover, analyzing organs at multiple time points can become very resource-intensive and will require a strong justification from an ethical perspective.

Probing a causal link between the formation of a pre-metastatic niche by a primary tumor and subsequent cell-specific homing was previously carried out in a challenging experiment [13]. In this setup, a primary tumor was implanted, which either secreted possible pre-metastatic niche-forming factors but did not metastasize itself or was resected before metastases could occur. In a second step, a metastasizing tumor cell line was injected into the bloodstream, and the metastasis pattern was analyzed against a non-secreting primary tumor control. Any differences could be attributed to the primary tumor and its niche-forming capabilities. This approach can enable the functional characterization of pre-metastatic mechanisms and by cross-linking different cell lines, it can describe ubiquitous niches across tumor entities. It is necessary to have a good understanding of the time points at which the primary tumor metastasizes, which is why this approach is probably limited to well-established tumor models. As is the case with any possible approach to a proposed pre-metastatic niche, a tumor model with reliable spontaneous metastasis is required.

### 5.2. Further Investigation Strategies for the PMN in the Brain

The exploration of pre-metastatic changes in the brain is challenging due to difficulties in obtaining spontaneous CNS metastases. In most cases, this necessitates the generation of brain-tropic cell lines (BrM) from existing tumor models. To obtain these specific cell lines, multiple rounds of tumor cell inoculation and in vivo selection from positive brains are performed [100]. The generated cell lines have the potential to reliably generate brain metastasis, but one could argue that the biology then becomes highly selected and diverges too far from the original tumor. To highlight specific differences between the parental and BrM cell lines, a comprehensive experimental setup should include the parallel cells. A comprehensive list of available selected cell lines is available online [101], rendering the first endeavors into the realm of in vivo brain metastatic research more approachable.

Arguably, the most restricting factor in studying pre-metastatic mechanisms in the brain is defining a suitable time point during the metastasis process and keeping the process as close to the natural biology as possible. As brain metastases are not as abundant in the course of the disease as visceral filliae, the humane endpoint of any experiment might be reached before meaningful conclusions can be drawn. To get ahead of the disease, approaches detecting the earliest micrometastases are needed (Figure 3).

The most pragmatic solution involves tumor inoculation/injection and the temporally spaced sacrifice of recipient animals, followed by measurements of the resident tumor cells in the brain tissues. This was performed by thin-slicing tissues and histopathological analyses [102]. Plotting the metastatic cells found in the tissue over these time points identified the first temporal occurrence of micrometastasis. This allows for the prior time point to be defined as the pre-metastatic niche. The benefits of this approach are the spatial-temporal reconstruction of the pre-metastatic niche and the identification of histological hotspots for tumor extravasation. On the contrary, biological variances in tumor growth and the metastasis process will make a clear division between a true pre-metastatic setting and overt metastasis difficult.

Diverse methods have been described to define the pre-metastatic niche in individual animals to make the results more consistent. Recently, it was shown that mCherry transcripts of subdermally inoculated mCherry-expressing melanoma cells could be measured in the CSF of tumor bearing mice and were an accurate predictor of incipient micrometastases in a spontaneous metastasis model [58]. The clear advantage of this approach is the early detection of micro metastases in a living animal, which could not be rendered otherwise. Although technically elegant, the experiment will probably only work with a limited selection of tumor entities that shed cells into the CSF. In addition, one could argue that the timeframe of a pre-metastatic niche has passed when the detection of cells in the CSF is possible, as the BBB must have then been breached already.

Intravital imaging has been fruitful in recent times and with the advancement of technology, the visualization of tumor cells in multiple organs in vivo has become possible [103]. The fixed position of the brain inside the skull also allows for methods not otherwise possible in visceral organs. Through a cranial window, blood vessels and the fate of individual tumor cells can be traced with great precision over multiple weeks. This enables a very precise in vivo depiction of cellular metastasis, although it is restricted to superficial metastasis. Furthermore, fruitful results can only be expected from experimental metastasis models (vascular tumor cell injection) as spontaneous metastasis models are too slow and unreliable [99,104,105]. The method is probably also limited in its widespread use as technical requirements are high and surgical expertise is necessary.

Advanced imaging techniques can be used to visualize aspects of the pre-metastatic niche. Emerging metastases can be detected by tissue-wide or whole-body in vivo luminescence and fluorescence, with new detection methods allowing for the highly sensitive detection of metastases, even though the resolution might not suffice for single cells or micro metastases [106,107,108,109,110,111]. Moreover, sophisticated tissue-clearing methods, combined with state-of-the-art open-top light-sheet microscopy for the rapid volumetric imaging of whole organs, has the potential to revolutionize the field [112,113,114,115]. Today, it is possible to clear and stain whole mouse bodies and create a pan-optic visualization down to single-cell resolution [116,117,118]. Additionally, expansion microscopy, the water-based polymer swelling of organs, can push histological tissue analysis to subcellular resolution with standard equipment [119,120,121]. Expanding on these imaging modalities, SPECT (single-photon-emission computed tomography)-based methods allow for the differential tracking of tumor cells and macromolecules in vivo and have been used to characterize tumor–immune interactions in a pre-metastatic setting [122,123,124,125]. Similarly, the method of choice to non-invasively image metastatic growths in vivo is high field magnetic resonance imaging (MRI) [106,110]. Here, ferritin heavy chain (FH1) tagging allows for the imaging of exosomes and other subcellular structures in pre-metastatic and metastatic niches [126]. Adapting these imaging methods opens up new dimensions for researching the early phases of the metastasis process. 

Beyond imaging, next generation and single-cell sequencing have equipped researchers with powerful tools that can be used to understand complex cellular interactions. Both allow for the analysis of individual cells rather than bulk tissue. This provides a comprehensive view of the cellular compositions and molecular signatures of metastases and distant organ sites. As the PMN and MN are composed of a diverse range of cells, each with their own unique molecular profiles and functions, it is especially important to be able to work with high spatial and temporal resolutions [127,128,129]. Contrasting the high resolution of novel sequencing techniques are technical shortcomings that still limit data interpretation. For instance, a single sequencing run can only display a momentary snapshot of the expression profile and could possibly miss crucial abnormalities. In addition, the characterization of the genotype alone would probably not suffice to describe highly specialized microenvironments such as the pre-metastatic niche. In this regard, so called multiomic approaches have recently been established, combining transcriptional sequencing with protein and protein–interaction analyses [130]. Similarly, in a technological push to combine imaging and genomics, MERFISH, Visium and GeoMX now position relevant molecular signatures into spatial contexts, a process known as spatial transcriptomics [131,132,133,134]. Applying these revolutionary methods can provide unprecedented insights into the individual steps of the brain metastatic cascade, especially with the development of high-throughput methods that can uncover genetic processes even in fixed pathology samples and circulating tumor cells [130,135,136]. The possibility of cutting edge RNA sequencing penetrating even paraformaldehyde-fixed samples allows researchers to utilize large databases of patient samples that have been accumulated in pathology departments [136].

### 5.3. Conclusive Remarks

The experimental approaches outlined in this manuscript, applied in symphony with each other, will aid in gaining a more comprehensive understanding of the metastatic and pre-metastatic niches and their role in the development and progression of brain metastasis. The elusive nature of the pre-metastatic niche, especially in the cerebral environment, asks for sophisticated and well validated methods. The technological advancements discussed herein, namely, experimental high-resolution microscopy and multiomic approaches, represent powerful tools that are critical for the advancement of metastatic research. Given that no single method of investigation will resolve the pre-metastatic niche in both spatial and temporal dimensions simultaneously, an interdisciplinary effort to integrate evolving methods is essential. This will identify novel drug targets for bench-to-bedside clinical trials.

## Figures and Tables

**Figure 1 ijms-24-10055-f001:**
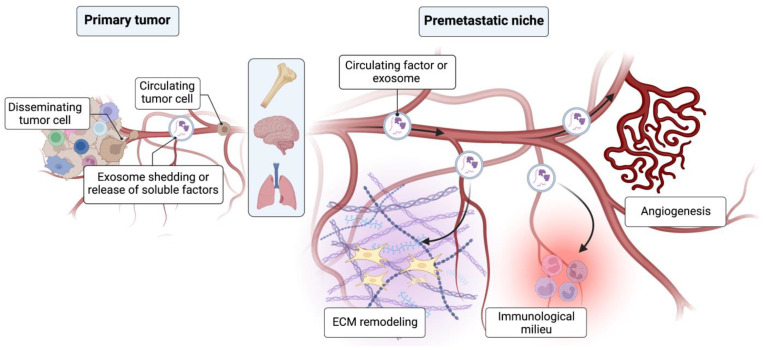
Concept drawing of the pre-metastatic niche. The primary tumor releases signals via exosomes and soluble factors to distant organs, preparing them for metastasis. Most of the distant modulation takes place in and around the microvasculature, with the endothelium being the first barrier for micrometastatic seeding. Among the most important aspects that define the pre-metastatic niche are the remodeling of the ECM, pioneering angiogenesis, the direct modulation of the endothelium and influences on the local immunological milieu.

**Figure 2 ijms-24-10055-f002:**
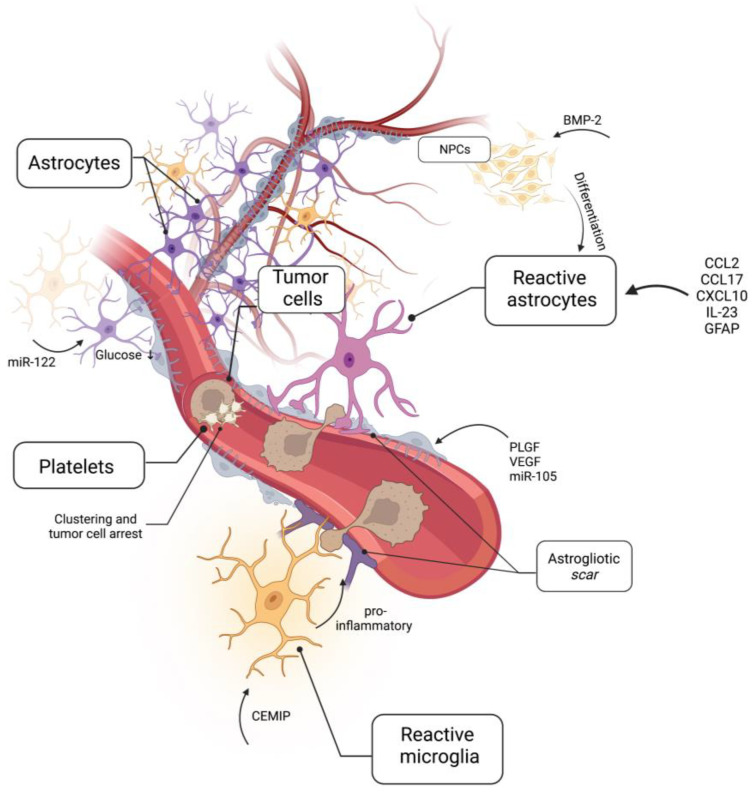
The blood–brain barrier in the brain PMN. The blood–brain barrier (BBB) is a unique feature of the brain and is virtually the only entrance site for circulating tumor cells. As such, it has unique gatekeeping properties that are routinely modulated and hijacked by primary tumors. What little is known about the brain pre-metastatic niche overwhelmingly concerns the BBB. Dissolution of the tight junctions by immunological factors and growth factors has been shown in a pre-metastatic setting. Astrogliosis plays an important role in pro-metastatic vascular leakiness, and multiple routes lead to it; inflammation of surrounding astrocytes leads to scarring and increased BBB permeability, as does the differentiation of neural progenitor cells into astrocytes.

**Figure 3 ijms-24-10055-f003:**
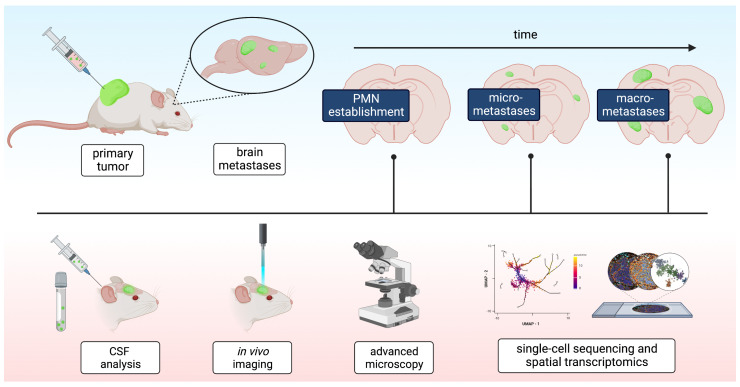
Advanced and experimental techniques for characterizing the pre-metastatic niche in the brain. In order to adequately describe the brain pre-metastatic niche, new and emerging techniques should be adapted to gain more insights into this elusive concept. In a spontaneous metastasis model, micro- and later oligmetastases will form in the brain. The timepoint right before the formation of the first micrometastases will represent the pre-metastatic niche most closely. Techniques such as elaborate CSF analysis for tumor cells or in vivo imaging of the brain could distinguish this timeframe. In a second phase, advanced microscopy methods combined with single-cell sequencing and spatial transcriptomics can be used to define it in more detail.

**Table 1 ijms-24-10055-t001:** Comparison of the pre-metastatic niche (PMN) and the metastatic niche (MN).

	PMN	MN
Function [7,8]	Access, anchorage, and early survival	Survival, protection, and proliferation
Effects [7,8]	Angiogenesis; Lymphangiogenesis; Vascular leakiness; Remodeling of the ECM; Inflammatory response.	Metabolic changes in the TME; Pro-survival signaling in the ECM; Induction of chemoresistance; Growth factor production; Early angiogenesis; Epithelial–mesenchymal-/mesenchymal–epithelial-transitions.
Exosomes	Transport of ncRNA at a distance [39,49,51,86]; Transport of proteins such as CEMIP and MIF [25,63]; Organ specificity via integrin expression [40];	Exosomal disposal of tumor-supressor miRNA in metastatic cells [44]; Paracrine ncRNA transfer [45].
ncRNA	*miR-21*: Angiogenesis upregulation through STAT3 [46]; *miR-105*: destruction of tight junction protein ZO-1 [19]; *miR-122*: modulation of glucose metabolism in stromal cells to favor metastatic cells [83]; *lnc-MMP2-2*: miRNA sponge, mitigating tumor-suppressive RNAs to increase BBB permeability [81]; *TLR3* upregulation and neutrophil infiltration in alveor epithelium by exosomal RNA [18].	*miR-19a*: *PTEN* downregulation in tumor cells, secreted by astrocytes [87]; *miR-135b*: secreted by hypoxic Multiple Myeloma cells to form endothelial tubes [45].
Inflammatory response	Decrease in *IFN-γ* and increase in pro-inflammatory cytokines in pre-metastatic lungs by immature myeloid cells [17]; *CXCR2* binding of chemokines might direct immunosuppressive MDSCs to the pre-metastatic site [37,88,89]; *S100A8/9* signal to accumulate macrophages in the pre-metastatic lung [90]; TAM recruitment by tissue-factor mediated coagulation has been discussed to take place even before metastatic cell arrival [91].	Direct cytokine regulation between tumor cells and astrocytes [92]; *IL-23* upregulation in metastasis-associated astrocytes [59]; *CCL2*-recruited TAMs promote metastatic outgrowth and angiogenesis [93,94]; Immune escape through T-cell suppression at the early metastatic site [28]; Immune shielding through platelet activation [95,96,97]; TAM recruitment by tissue-factor mediated coagulation promotes metastatic survival [91]; In vitro data regarding T_reg_ cells in the MN (discussed in [37]).
Vascular response	*miR-105*: modulates brain endothelial tight junctions [19]; Endothelial cell remodeling through *STAT3*-highjacking [20,46]; Remodeling through *Notch1*-receptor upregulation [22]; Remodeling through *MMP9* [17].	Tumor-cell-mediated necroptosis of endothelial cells [98]; Vascular cooption in brain vessels [71,99]; Interplay between microvasculature and dormant tumor cells [7,27].

## Data Availability

No new data were created or analyzed in this study. Data sharing is not applicable to this article.

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
