# Peer review of "The Brain Pre-Metastatic Niche: Biological and Technical Advancements"

_ijms, 2023, doi:10.3390/ijms241210055_

Round 1
Reviewer 1 Report
The review titled 'The brain pre-metastatic niche: biological and technical advancements' provides a comprehensive overview of the pre-metastatic and metastatic niches with a specific focus on the brain microenvironment and factors affecting metastasis to the brain. The authors have done a commendable job of compiling recent literature in the field and presenting the current understanding of the field. Additionally, the authors attempt to highlight the existing gap in our knowledge and provide some details with regard to future approaches which might help fill the existing gaps and further refine our understanding of PMNs and MNs. Overall, the review is well-written, comprehensive, and systematically presented. The authors do provide some supportive figures and tables to better support the presented information and highlight the important concepts to the reader. The review is refined and of significant importance to the field. The few concerns associated with the manuscript have been listed below. The objective is to make the review more reader-friendly and further improve the quality of the presented information.
1) The presented figures (figure 1 and 2) provide a very general overview of the concepts. The issue then is many previous reviews have presented similar figures to explain the same concept. The authors are encouraged to associate their figures with the specific ideas presented in the review. It might be worth the effort to provide details that align with the review literature and provide either specific examples of verified data that fit within the figure. While it is true that the manuscript is a review, the focus of the manuscript is lost in presenting general abstract figures.
2) The authors have seldom discussed the caveats or shortcomings associated with future investigation strategies. It is important to present the reader with an unbiased view of how these future strategies still need improvement and could present an experimental or systemic bias when interpreting results. It is also important to provide technical challenges associated with future investigation strategies.
3) The final discussion sections (last two paragraphs) within investigation strategies need additional work. The authors have listed the techniques which might help better understand PMNs. However, the authors have assumed that the readers are familiar with the working details of these techniques. The authors need to provide brief details on how each listed technique works and could help understand some aspects associated with PMNs. The authors also generalize the processes that could be investigated using these techniques. The authors can definitely better breakdown the merits and demerits of each technique and associate them with potential future experimental questions.
Author Response
"Please see the attachment."

Reviewer 2 Report
I found the topic of this review to be interesting. The introduction is excessively general and lengthy. The topic of the paper cannot be an overview, but must be specific and indicate the contribution and the innovations it introduces. There is no section explaining the methodology used in carrying out the review, or the criteria for inclusion of articles, in this section. I understand that the division into six sections is unusual, please convert them into a discussion. Furthermore, further investigation strategies for the PMN in the brain should be included in the discussion and replaced by a conclusion section. What is deduced from the literature must be clearly explained and schematized, and the novelties introduced by the revision and its originality must be highlighted. Please conduct an extensive review and explain the methodology used for the review and outline the report. Also, please indicate whether Figures 1 and 2 were produced by the authors or any other source.Author Response
"Please see the attachment."

Round 2
Reviewer 2 Report
In this review, we aim to present recent findings about the brain pre-metastastic niche and discuss existing and emerging methods to further explore the field. We begin by providing an overview of the pre-metastatic and metastatic niche in general, before focusing on their manifestations in the brain. The article, however, cannot be accepted in this form. I think it would be helpful to add a section explaining the methodology for selecting the articles cited in the review. I feel that sections 2, 3 and 4 are too general and too wordy. In the discussion section, the changes introduced by this revision should be better emphasized. The conclusions should focus almost exclusively on the novelties that have emerged from this review. Figures 1, 2, and 3 should be indicated whether they were made by the authors or if they contain a quotation from which they originated. Many of these comments were already suggested in the previous revision, and have not been introduced yet.
Author Response
"Please see the attachment."

Round 3
Reviewer 2 Report
After the changes made the article can be accepted.